# A results to action framework for community verification: A case study from a performance based financing program in Zimbabwe

Trina Gorman[1,2]*, Bernardo Hernandez[3,4], Geoff Garnett[5], Loida Erhard[2], Taurai Kambeu[6], Malvern Munjoma[6], Brian Maponga[6], Sinokuthemba Xaba[7], Getrude Ncube[7], Jabulani Mavudze[6], Gabrielle O'Malley[1]

1 Department of Global Health, University of Washington, Seattle, Washington, United States of America, 2 Gorman Consulting, Edmonds, Washington, United States of America, 3 School of Public Health/National Institute of Public Health of Mexico, Cuernavaca, Mexico, 4 Department of Health Metrics Sciences, University of Washington, Seattle, Washington, United States of America, 5 The Bill & Melinda Gates Foundation, Seattle, Washington, United States of America, 6 Population Services International, Harare, Zimbabwe, 7 The Ministry of Health and Child Care, Harare, Zimbabwe

* trina@gormanconsulting.org

## Abstract

Performance-based financing (PBF) is a funding strategy that pays for outcomes rather than the cost of inputs. Verification through facility records (quantity verification) and patient interviews in communities (community verification) is a known cornerstone of PBF to ensure reported results are accurate. However, the literature suggests it's common to tie payment to quantity verification results, which measure internal record alignment but do not assess the validity of records (e.g., whether records represent delivered services). We sought to understand the extent to which reported voluntary medical male circumcisions (VMMCs) in a PBF program could be verified in facility records and with patients, and if the two sources aligned at the facility-level. We performed a mixed method verification including quantity verification and community verification to verify reported results for Population Services International's VMMC program in Zimbabwe from 2016 – 2018. We also interviewed verifiers to help understand the findings and we assessed the correlation between quantity and community verification performance scores at the facility-level to see whether facilities that have strong record keeping tended to also have strong validation from patients and vice versa. Among the 36,877 VMMCs selected from DHIS2 for quantity verification, 94% of records were sufficiently complete. Among records selected for community verification, only 55% (2,010/3,676) of patients were interviewed. Among those interviewed, 17% (342/2,010) provided answers that did not plausibly match the record. Verifiers reported that some patients admitted providing incorrect contact information to avoid follow-up and most verifiers suspected staff had fabricated data. We found no correlation between performance scores at the facility-level. Overall, results from the quantity verification were not a good proxy for the community

**Data availability statement:** There are ethical or legal restrictions which prevent the publication of minimal data for this study. The minimal data contain personal information which may compromise the privacy of participants. Data are available upon request from Aleck Dhliwayo, MIS manager, via email (adhliwayo@psh.org.zw) for researchers who meet the criteria for access to confidential data.

**Funding:** The authors received no specific funding for this work.

**Competing interests:** The authors have read the journal's policy and have the following competing interests: Geoff Garnett is a paid employee of The Gates Foundation. This does not alter our adherence to PLOS ONE policies on sharing data and materials.

verification. Programs that pay based on facility records alone risk overpaying for services and misreporting performance. To increase the use of community verification findings, PBF programs should consider using and improving our proposed results to action framework.

## Introduction

HIV/AIDS is the leading cause of death in Zimbabwe [1], totaling 20,000 deaths per year [2]. Substantial resources have been dedicated to HIV/AIDS prevention, including the expansion of Voluntary Medical Male Circumcision (VMMC) to reduce transmission. The Ministry of Health and Child Care (MoHCC) has leveraged Performance-Based Financing (PBF) as part of its VMMC strategy. PBF is payment for pre-determined, verified outcomes rather than payment for inputs [3]—a funding approach that has been increasingly implemented in low and middle income countries, totaling over $19 billion in funding since 2000 [4]. The theory that underpins PBF is that aligning provider payment with provided services increases health care worker productivity and service quality [5,6].

Despite potential advantages of PBF schemes, they are continually challenged by weaknesses in the health management information systems (HMIS) that track reported outcomes as well as by the potential for perverse effects (e.g., fraud, gaming, etc.) [7–9]. Independent verification is critical to ensure reported outcomes, payment, and the perceived health benefit reflect reality [10,11]. Verification activities aim to identify and reduce overreporting, which can be either intentional (e.g., workers fabricating data to increase pay) or unintentional (e.g., bookkeeping errors due to insufficient time or training). Typically, PBF verifications include reviews of paper-based facility records (also called quantity verification) to ensure records are complete and interviews with patients in communities (called community verification, client tracing) to ensure records represent delivered services since records could be fabricated [10,11].

While the explicit purpose of community verification is to determine the validity [12] of the facility records, the literature suggests the results from these efforts are often neglected both for programmatic decision making and in the literature. A review identified ten PBF verifications that include quantitative results, including eight from the grey literature [13–20] and two that were peer reviewed [11,21]. Only seven of the ten articles include community verification [11,14–16,19–21] and of those that did, three explicitly report that the results were not used to hold actors accountable for overreporting or fraud [11,15,21]. This implies that in six out of the ten verification studies, payments were tied to records alone, which is counter to PBF guidance that verifications should "signal to providers that there is a strong chance that one will be caught if one cheats (by claiming phantom patients)" [10]. Tying payment to records alone would be less of a concern if there were empirical evidence that facilities having strong records tended to also have strong community verification results and vice versa. However, we found no studies that assessed the correlation of these two performance scores.

In addition, the available literature provides little detail on the methods and findings of community verification. Most studies only contain high level quantitative results without details regarding indicator definitions or the processes that led to the results; as an example, only three [14,19,21] of the seven studies [11,14–16,19–21] reported on both the portion of selected patients that were interviewed and the portion of interviewed patients whose responses matched records.

Measuring the extent to which facility records represent delivered services is essential for the effective implementation of PBF. However, the literature is scant regarding community verification results and the paths and processes that led to them. This study helps to fill this gap by describing a quantity and community verification for a program designed to encourage scaling VMMC in Zimbabwe, including the reasons behind the results from the view of the verifiers.

## Methods

### Study setting

HIV in Zimbabwe has been a significant public health concern for many years. The incidence of HIV/AIDS in 2019 was 214 new infections per 100,000 people per year - a significant decrease from the 669 new cases per 100,000 people in 2009 [22]. The VMMC program in Zimbabwe launched in 2009, following randomized trials that showed circumcising men reduces the risk of female-to-male sexual transmission of HIV by up to 60% [23–25].

### PBF program design

Starting in 2015, the Bill & Melinda Gates Foundation supported Zimbabwe's VMMC program through a series of grants to Population Services International (PSI), in partnership with the MoHCC. This study focused on VMMCs performed from May 2016 to April 2019, whereby PSI as well as PSI's facility staff were remunerated for each VMMC; for staff, the incentivized payments were in addition to an unconditional base pay. Each VMMC patient was documented with one paper record at the facility. Initial payments to staff were made based on the number of paper records. If the quantity verification or community verification activities described below identified unvalidated VMMCs, then a staff's future payments were reduced based on a discount factor. If unvalidated VMMCs were deemed to be the result of fraud, the responsible staff were additionally dismissed. At the start of the program, staff in 12 health facilities in 12 districts performed VMMCs in their own facilities and on outreach at VMMC locations; the program was then decentralized to where 32 facilities performed VMMCs across 355 locations and 18 districts.

### Verification and study design

The verification was completed at four time points. At each time point verifiers were trained over three days; the verifiers included PSI staff, MoHCC staff, and contractors. For each time point, verification activities were performed between 1–12 months following when the VMMC was reportedly performed. Verifiers were paid a daily rate for their time.

**Quantity verification.** Verifiers visited health facilities to quantify the alignment between the number of 'sufficiently complete' facility records and the aggregated totals per month and facility that were stored in PSI's health management information system (DHIS2). To be classified as sufficiently complete, a facility record required the following fields: facility name, procedure date, patient name, patient date of birth, patient address, patient phone number, patient consent, whether HIV testing was offered, provider name, and provider signature. Verifiers recorded the verification results on a custom paper form based broadly on the July 2009 version of MEASURE Evaluation's Routine Data Quality Assessment tool [26] and then supervisors entered the results into Excel. A purposeful sample of records were selected to fulfill requirements from the funder, which was that 25% of records needed to be verified. At each of the four time points, PSI staff selected between 6–9 months of reported VMMCs, based on what was needed to be on track for the 25% requirement. The sample focused on facilities that reported the largest numbers of VMMCs; 41% of VMMC locations (144/355) were selected at least once. Once selected, a census of the facility records that fell into the time period

were reviewed. To complete this work, data from facility records was accessed over the following periods: 17/09/2016–11/11/2016; 12/04/2017–23/6/2017; 7/6/2018–01/09/2018; 05/04/2019–28/05/2019.

**Community verification.** Among facility records that were sufficiently complete, 10% were randomly selected for patient interviews in communities (2.5% of reported VMMCs). If patients were unavailable after three attempts of visiting the community, they were marked as unavailable. Patients or their guardians (in the case of minors under 16 years) were interviewed on the phone if possible or at their homes; responses were captured on a mobile phone. They were asked if they were circumcised, and if they were, they were asked three additional questions about the procedure, i.e., the general time period, district, and VMMC method (surgery or prepex device). If their response from one or more of these questions did not plausibly align with data captured from the record into Excel, the VMMC was considered unverified.

To learn more about why some patients whose records were selected for community verification could not be interviewed, verifiers who had performed the community verifications were recruited and subsequently interviewed between 01/09/2020–01/10/2020. Among all verifiers for the community verification (Table 1), those who performed less than 10 patient interviews and who only worked during the first verification were excluded given program changes and recall bias concerns. Interviews were conducted in English over Zoom using a standardized semi-structured questionnaire. Verifiers were asked to provide Likert scale rankings regarding how frequently they encountered certain scenarios. They also ranked their certainty level regarding whether patients or VMMC staff had fabricated data, where 0 meant they had no suspicion and 10 meant they were certain. Verifiers were also asked open-ended questions to explain their rankings. Responses were transcribed in real time into an electronic survey and audio recorded; the recording was used to complete the transcription following the interview. Additional information regarding the ethical, cultural, and scientific considerations specific to inclusivity in global research is included in the Supporting Information (S1 Checklist).

## Ethics statement

The protocol was approved by the research council of Zimbabwe (approval #MRCZ/E/237). The University of Washington's human subject's department classified the research as exempt. Patients who were interviewed provided oral consent and verifiers that were interviewed provided written consent.

## Analysis

Descriptive results are presented for the quantity and community verification. To assess the correlation between the two performance scores, the percentage difference between the verified VMMCs (e.g., sufficiently complete facility records) and reported VMMCs was calculated for each VMMC location using the first quantity verification results; the same was calculated using the community verification results. Pearson correlation coefficients were then calculated along with a univariate linear regression model to assess the linear association between the two results. Each result was further defined using a binary measure of overreporting if the reported number of VMMCs exceeded the verified number of VMMCs by 10% or more, a standard in the field [27,28].

Close-ended questions were summarized using Stata (StataCorp. 2021. Stata Statistical Software: Release 17. College Station, TX: StataCorp LLC) and qualitative responses were coded using Dedoose software (Sociocultural Research

**Table 1. Verifiers for the community verification.**

| Verification time point | Number of verifiers |
| --- | --- |
| 1 | 23 |
| 2 | 40 |
| 3 | 31 |
| 4 | 28 |

Consultants LLC, Los Angeles, CA) through both deductive and inductive approaches. An initial codebook was developed using a parent code for two general areas of inquiry: reasons why patients were not found and reasons why patient responses did not match the facility data for the four required questions. Nested within these codes, each survey question was assigned a child code and additional grandchild codes were developed inductively through consensus between two coders.

## Results

Fig 1 summarizes the results from the quantity and community verification. For the quantity verification, 94% (34,713/36,877) of selected facility records were sufficiently complete (C). Of the 3,676 patient records selected for

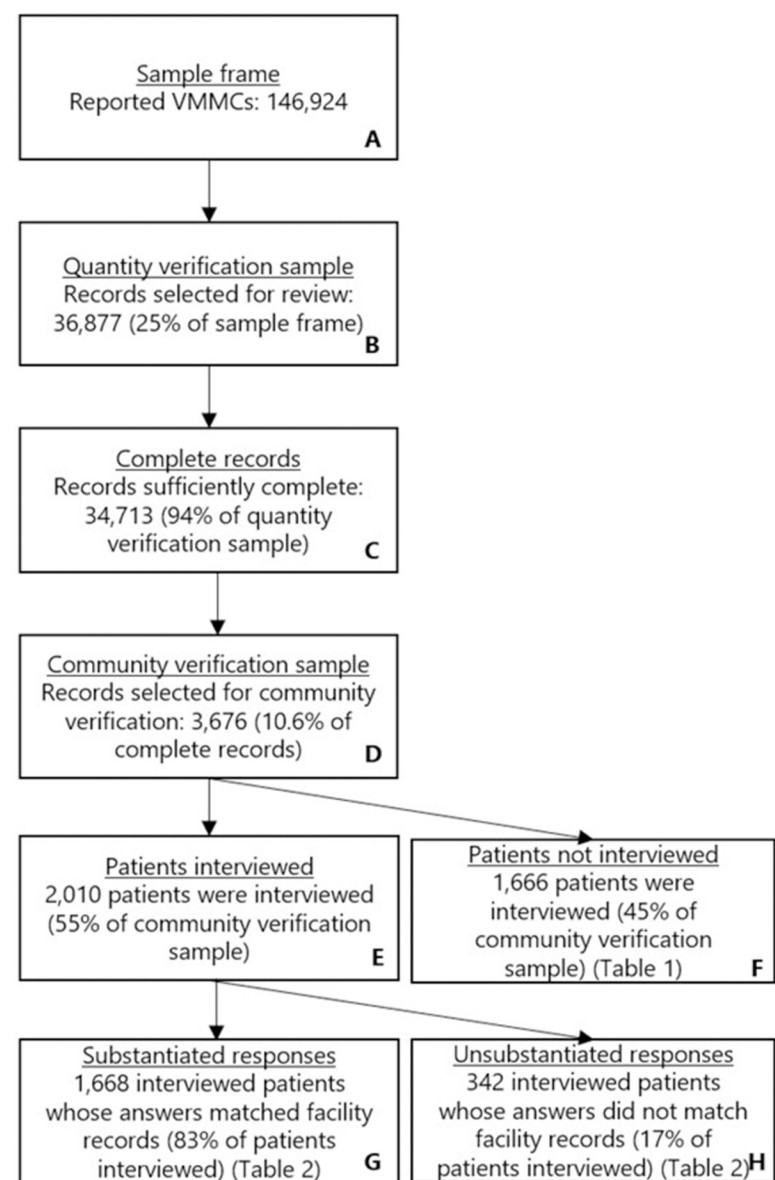

**Fig 1. Cascade of sample and results.**

community verification (Table 2), community verifiers were only able to interview 55% (2,010) of patients (E). Table 3 summarizes the reasons patients were not interviewed, including that 18% were marked as both unknown and had either insufficient or invalid contact information. Among interviewed patients, 83% (1,668/2010) met the funders requirements to verify the VMMC (G).

In terms of the correlation of the performance scores at the facility-level, we found no statistically significant correlation based on both the Pearson correlation coefficient (r=-.034, p=0.68) and linear regression (β=-.018, p=0.69). When instead using a binary measure of overreporting with a 10% cutoff, 43% (58/136) of results were aligned (either both over-reporting or both not overreporting).

For the verifier interviews, 34 verifiers were interviewed among 37 invited. Those interviewed had completed 76% (1536/2010) of the community verifications. In the rest of this section, we summarize why patients were not interviewed (Step F) and why patient responses did not plausibly verify the VMMC (Step H) from the views of the verifiers.

### Reasons patients were not interviewed

Verifiers highlighted three main reasons why they thought such a large portion (35%) of patients were reported as unknown in their community (Table 3): suspected falsification of data, unintentional bookkeeping errors, and verification design decisions.

**Table 2. Description of patients selected for community verification.**

| Description | Mean (SD) or Proportion | N |
|---|---|---|
| **Characteristics of patients** | | |
| Age of patient in years | 17 (0.55) | 3676 |
| *Method of circumcision* | | |
| prepex | 8% | 3623* |
| surgical | 92% | 3623* |
| **Characteristics of patient's VMMC location** | | |
| *Type of facility* | | |
| district | 25% | 3676 |
| mission | 19% | 3676 |
| rural clinic | 46% | 3676 |
| rural hospital | 10% | 3676 |
| *Type of base funding* | | |
| council | 27% | 3659* |
| government | 48% | 3659* |
| mission | 24% | 3659* |
| private | 1% | 3659* |

**Table 3. Summary of reasons patients could not be interviewed.**

| Reason Patient was not interviewed | | n | Percent of all selected patients |
|---|---|---|---|
| Known in community | … but not available (working, relocated, passed away) | 250 | 7% |
| Unknown in community | … and insufficient contact information | 503 | 14% |
| | … and sufficient contact information | 495 | 13% |
| | … invalid address | 143 | 4% |
| | Without further explanation | 160 | 4% |
| Other | Unspecified | 28 | 1% |
| | Lost interview data | 87 | 2% |
| Total | | 1666 | 45% |

**Suspected falsification of data.** Nearly all verifiers suspected that at least some data on the facility records had been made up by either staff or patients. When asked to rank their certainty level regarding *who* had provided false data, Fig 2 summarizes their responses.

Nearly half of verifiers reported that patients had admitted providing false/misleading contact information (e.g., nicknames, uncle's name) because they didn't want to be followed-up; their reasons included social stigma, desire for privacy, or because a minor had been circumcised without his parent's consent.

*I met quite a number of beneficiaries and having a chat with them, they would explain "when I was provided the VMMC service, I actually used the wrong name." The first name was right, but the surname was wrong or vice versa. (ID 109)*

*In the scenario of students, they were mobilized in the school for circumcision. Then they would go home with the consent form and parents would disagree for them to participate. The kids would sign their own forms with false names, so their parents don't know. (ID138)*

Most enumerators reported having some suspicion that clinicians or mobilizers fabricated data. Verifiers reported their suspicion was primarily due to experiences they had during the field work. For example, some verifiers described seeing questionable facility records while extracting patient information for the community verification (e.g., fields that appeared to have been duplicated, repeated patient signatures, or different ink types).

*You would find that on some CIRs [clinical intake form] you would see that even the consent forms have been signed by one person, sort of forged the signature. You would see the information is false [because] the same signature was on multiple CIRs. (ID 132)*

*The information was similar to another CIR that you looked at. Especially when doing the CIR review - people would call each other and say "it appears that the information was copied from another CIR into another CIR". (ID 121)*

Another reason for suspicion was invalid or conflicting contact information from the facility record, such as when the address was an area without dwellings (bus stop, demolished houses) or when the plot number went beyond the possible range.

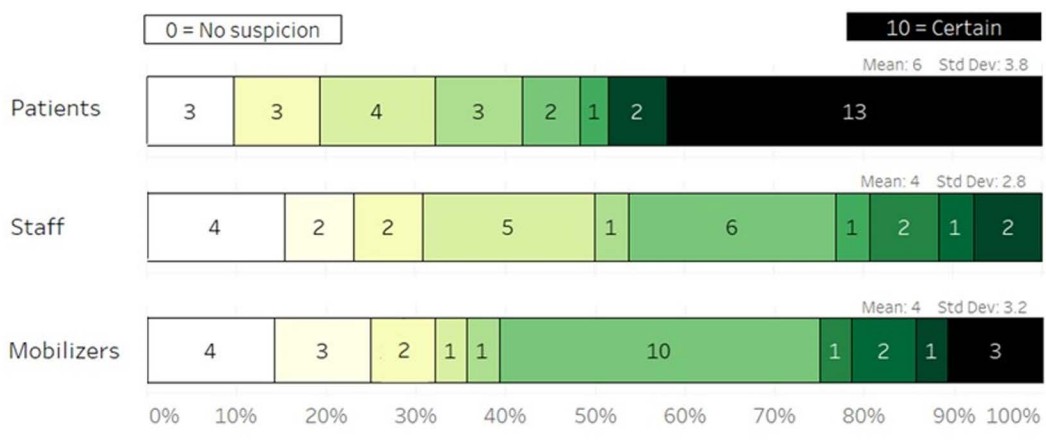

**Fig 2. Verifier's certainty level of falsified data, per group.**

*"This area was demolished (old houses in the colliery) so everyone knows that no one stays in those addresses anymore. But you would find these addresses being recorded as successful male circumcisions. Why would you record this as an address when [you know it] is no longer in use?" (ID 128)*

*"Like the address 21 Harare Street. It's impossible to have a street in such a rural area." (ID 126)*

Staff behavior was also mentioned as an underlying cause of suspicion, such as when mobilizers were not able to help find patients they had reportedly previously mobilized. Four verifiers said a mobilizer intentionally brought them to the wrong patient to try to manipulate the verification results; three verifiers said they saw nurses filling in facility records for VMMCs when they arrived for the quantity verification though there were no VMMC patients at the time.

*If you are moving around a village, at least you should be able to locate 70–80% of the people. Once you start going around and no one knows the name or surnames, you start wondering where the mobilizer got the information from to get them for circumcision as they are so busy and the walking distances are so far. (ID 113)*

Verifiers said they expected to not find some patients, but many said that the large proportion of unknown patients made them suspicious data had been fabricated. On average, verifiers expected that 15% of the time village leaders and other contacts would not know a person in their village, in contrast to the 35% of patients that were reported as not known in this study.

*I was suspicious when we would talk to all the people who should know in a village set up - they have homes close together and should know each other. We would visit the shops etc. and the people are sure the person isn't there. They would refer to someone who knows everyone in the village, and they would say there is no one here by that name. Then we would follow up and ask the mobilizer - he would say "I think he relocated" but it wasn't convincing. (ID 106)*

For other verifiers, it wasn't only their experiences from the fieldwork that made them suspicious but also their belief that the financial incentive coupled with economic hardships and unrealistic targets created too great of a temptation.

*From the home visits and checking their attitude, I would say they were really chasing targets because they are paid per client. So to me, looking at the economic situation they were out there to make money and those clients are not easy to get…. (ID 119)*

Despite their suspicions, most verifiers admitted that they didn't have hard evidence about fabrication by staff.

*We didn't have tangible evidence - we didn't have mechanisms to follow-up to investigate more on these instances. But we had very strong suspicions... we felt, if something was done - like an investigation - there was a high chance the conclusion or finding would point to instances where the teams had been fixing some figures (ID 121)*

However, three verifiers said they had no suspicion of fabrication by either clinicians or mobilizers.

*I don't think they provided wrong data because whenever we find them, they would take you straight to the client's home. … These guys they really knew exactly where the client lives or they would point you in the right direction. (ID 104)*

**Unintentional bookkeeping errors.** Several verifiers believed that some missing patients were due to unintentional bookkeeping errors, exacerbated by high workloads. A few verifiers recalled asking staff why so many patients could not be found, which helped reduce suspicion for some verifiers but not others.

I had to confront the nurses that were responsible because we were failing to locate the clients. She provided insight that maybe at times they were under too much pressure during the day with the VMMC procedures that they wouldn't fill in the documentation…. So, the [reason for the] funny CIR is that the client did exist, but they just failed to fill in the details that would be required. But I still do believe there was some funny stuff going on – they aren't busy all the time with outreach. (ID 114)

**Verification design decisions.** Most verifiers said a main cause of not locating patients was a lack of detailed contact information since the program did not require specific addresses; at times a large village or school was all that was listed. Some verifiers mentioned more patients could have been found with more transportation funds and time for the field work.

### Reasons interview responses did not plausibly match the facility record

7.5% of interviewed patients (151/2010) were self-reported as not circumcised. Verifiers said these 151 patients commonly reported they had registered for the VMMC but then did not complete the procedure. When verifiers were asked how they thought a complete record was found at the facility for these patients, they explained that it was common for staff to pre-fill records with basic information about patients, leaving just a few fields for afterward. Verifiers were generally split regarding whether they thought the remaining fields on these uncircumcised patients' records were intentionally fabricated or filled in by mistake.

9.5% of interviewed patients (191/2010) reported being circumcised but provided an implausible answer regarding the time period, district, or method, as shown in the grey cells of Table 4; Table 4 summarizes the responses from interviewed patients, excluding the 151 that reported not being circumcised. 114 patients reported the facility and method plausibly, but not the date; 58 patients reported the facility and date plausibly but not the method. To learn about these 58 patients, verifiers were asked about the survey question that asks patients about the method used in their procedure; nearly one quarter of verifiers thought that patients at times may not have understood the difference between the two methods even after verifiers provided additional explanations. Finally, it is also possible that some verifiers interviewed the wrong patients: when asked whether they completed a survey even when they were not certain that the respondent was the right person, most said never, while four verifiers said rarely or sometimes.

### Discussion and program recommendations

To our knowledge, this is the first mixed method study examining a quantity and community verification with detailed findings including the paths and processes that led to them. Overall, we found that quantity verification results were not a good proxy for community verification results for two reasons. First, while the quantity verification results were high (94%), only 45% of participants were reachable for community verification interviews, and among those interviewed, 17% could not be validated. Second, we found no correlation between the two performance scores at the facility-level. The purpose of verification is to detect overreporting by staff and most verifiers did suspect some data fabrication in this regard;

**Table 4. Reported date, method, and facility from interviewed patients who reported being circumcised.**

| | | Time period not plausible? | | Time period plausible? | | |
| | | Method plausible? | | Method plausible? | | |
| | | No | Yes | No | Yes | Total |
|---|---|---|---|---|---|---|
| Clinic plausible? | No | 1 | 6 | 1 | 6 | 14 |
| | Yes | 5 | 114 | 58 | 1668 | 1845 |
| Total | | 6 | 120 | 59 | 1674 | 1859 |

however, verifiers also believed discrepancies were caused by others (PSI, patients, verifiers themselves) and they only had hard evidence about patients fabricating data. These findings support two points of discussion in the literature.

First, verifiers' suspicion that staff fabricated data is both supported by our previous quantitative analysis focused on factors associated with overreporting in this same program [29] and echoes evidence from a series of studies on the unintended consequences of PBF. Turcotte-Tremblay et al. performed in-depth case studies in Burkina Faso using data gathering techniques inspired by anthropology, such as living at facilities for two weeks, and found that in two [21,30] out of three studies [21,30,31], there was widespread data fabrication by staff paid per output in order to increase payments. In one study, staff in all six study facilities routinely spent considerable time creating documentation for unperformed services [30]. Another study found that when the verifiers tasked with verifying the patient data were paid per patient themselves, they also falsified data in the majority of cases using deliberate and organized strategies [21]. In the author's third study, staff still manipulated the program but instead to gain non-financial benefits [31].

In terms of understanding why community verification results are often not analyzed [14,32] or used to inform payment [11,15,21], our findings reinforce a primary reason reported by one of the Burkina Faso case studies [21]. The authors explain that multiple parties were responsible for data discrepancies, as the verifiers in our study also reported, which made it challenging to interpret the results and agree on appropriate actions [21]. For example, a discrepancy between a patient's response and facility data does not necessarily mean overreporting occurred [21]. While it could have been caused by staff who recycled patient data (overreporting), it could have instead been caused by a patient's memory lapse or a verifier who interviewed the wrong person (not overreporting, but measurement error).

Given these complexities, a framework may be useful to guide action based on community verification results. Mapping results to actionable recommendations has been identified as an important step in increasing the use of data for program improvement [33,34]. Table 5 presents a "Results to Action" framework, which organizes the type of result, responsible party, and potential underlying causes as found in our study and other studies [11,21,30,32,35]. Below we elaborate on possible follow-up actions, drawing on our findings, the available literature, and the PBF program theory.

Three primary recommendations are proposed in this framework for future research and program implementation. First, it is important that PBF programs reduce payment when a patient's response does not plausibly match the facility record. The theoretical foundation for PBF is that it helps solve the principal-agent problem through paying the agent contingent on verified outcomes—so that the principal pays only for what has been achieved [4,36,37]. The entire purpose of the verification is therefore to measure *achievements* [10]. If community verification results are not used, it weakens the power of the incentive [11]; there is virtually no way to identify cheating based on records alone [31]. When it is not possible to verify whether patients received services (e.g., due to logistical/financial constraints or patients that will not consent to follow-up for privacy/stigma reasons), PBF may not be optimal in that setting [10].

Second and in contrast, programs should be cautious about reducing payment for patients who are not found for interviews unless there are good estimates regarding the portion of errors for which patients and verification teams are likely to be responsible. While overreporting could be disguised as patients who were not found for interviews, our findings align with past studies showing it's common for verifiers to have insufficient resources to find all patients [11,21,32], and for patients to avoid follow-up [16] or be away when verifiers visit [19–21]. Reducing pay without sound estimates regarding these occurrences risks unfairly penalizing the program.

Third, programs need to make sure that verifiers produce data that is reliable by ensuring teams are well trained, paid sufficiently and on time, and have reasonable oversight [32]. Evidence from various fields including academia—from low and middle income countries [38–40] to high income countries [41,42]—has shown that data quality suffers when accountability and incentive structures are weak or perverse. Counter to this, the same Burkina Faso study mentioned earlier paid verifiers on a per patient basis only when patient responses matched the health facility data [21]; verifiers were also not paid if a patient was unavailable but had not permanently moved. Unsurprisingly, the verifiers fabricated data to increase their pay. Lack of adequate pay [11,21], training [32], and time for verifiers [21,32] has been reported by many community

**Table 5. The results to action framework for community verifications.**

| Type of result | Responsible party | Potential underlying cause | Possible follow-up actions | |
|---|---|---|---|---|
| | | | **How to avoid this underlying cause from occurring?** | **Apply sanctions if underlying cause is unknown?** |
| Patients not reachable for an interview | Health facility staff | Intentional data fabrication or fraud (e.g., ghost patients) | - Perform verifications and apply sanctions using the results<br>- Ensure sufficient staffing; pay and train staff well | Only when programs can estimate the portion of missing patients that are likely to be due to the patient and verification team, or when programs are confident that this portion is low. |
| | | Unintentional bookkeeping errors | | |
| | | Insufficient contact information collected from patients (e.g., typos, nicknames, vague addresses, patients with the same name, too few contact options) | - Collect detailed contact information & references | |
| | Patients | Insufficient contact information provided either intentionally (e.g., patients avoiding being found) or unintentionally (e.g., wrong phone number, nicknames, name changes over time) | - Avoid tying payment to services associated with stigma or services that are very personal | |
| | | Unavailability either unintentionally (e.g., transient populations/visitors, working class, moved) or intentionally (e.g., patients avoiding survey attempts) | - Avoid tying payment to services used by transient populations<br>- Schedule verification interviews when patients are likely to be home (e.g. not during working hours) | |
| | Verification team | Lack of effort to find patients | - Ensure sufficient staffing; pay and train verifiers well | |
| | | Insufficient resources (e.g., limited transportation funds, time restrictions) | - Invest sufficient resources | |
| Patients whose responses do not plausibly match health facility data | Health facility staff | Intentional data fabrication or fraud (e.g., recycling of data from real patients) | - Perform verifications and apply sanctions using the results<br>- Ensure sufficient staffing; pay and train staff well | Yes, this is imperative to uphold the accountability structure |
| | | Unintentional bookkeeping errors | | |
| | Patients | Lack of truth telling (e.g., stigma, concern for privacy, social desirability) | - Avoid tying payment to services associated with stigma or services that are very personal<br>- Ensure verifiers gain rapport | |
| | | Memory lapse | - Avoid tying PBF payment to services people are likely to forget<br>- Avoid selecting patients for verification that are not likely to remember the services (e.g the elderly) | |
| | | Misunderstanding the interview questions | - Avoid selecting patients for verification that are not likely to understand the questions (e.g the elderly); hire experienced verifiers; train/pilot thoroughly | |
| | Verification team | Data falsification (e.g. a patient's responses are fabricated) | - Ensure sufficient staffing; pay and train verifiers well | |
| | | Data entry errors when extracted from the facility | | |
| | | The wrong patients are interviewed | | |
| | | Questions are asked incorrectly | | |
| | | Data entry errors when patient responses are transcribed | | |
| | | Data from patients are incorrectly matched to data from facilities | | |

verifications. While there is increasing focus on avoiding perverse incentives for PBF staff [43–47], programs cannot lose sight of the importance of verifier incentives as well.

This study has several limitations. First, the selection of facilities for quantity verification was purposeful, and as a result, so were the patients selected for community verification. Most notably, facilities that performed fewer VMMCs were not included so the findings might not generalize to those areas. The low portion of patients that were interviewed is important to understand challenges with finding patients, and yet reduces the internal validity of findings from interviewed patients. When considering the data quality of the community verification, it is possible that the verifiers fabricated some surveys due to common pressures. However, verifiers were paid sufficiently and for their time (not per survey), and were not pressured to complete many surveys, so had low incentives to do so. Further, VMMC is a sensitive, stigmatized topic in some populations in Zimbabwe so findings might not translate to other programs where patients are more open to follow-up. Finally, interviews with the verifiers covered sensitive topics about data fabrication, so are subject to a set of biases. Of most concern was social desirability bias and recall bias, which we mitigated with a thorough informed consent, sequencing of topics, and repeatedly removing any pressure to answer.

## Conclusions

We found that when verifying reported VMMCs in a PBF program in Zimbabwe, the results based on the quantity verification (i.e., completeness of facility records) were not a good proxy for accuracy when compared to findings from the community verification regarding whether the services were actually delivered. While verifications are complex in that inaccurate results can be caused by many parties, reducing payment in cases when patient responses do not match facility records helps ensure programs are paying for *achieved* results and that there is a real threat of sanctioning for intentional overreporting—two core theoretical requirements for PBF.

## Supporting information

**S1 Checklist.  Inclusivity in global research.**
(DOCX)

## Author contributions

**Conceptualization:** Trina Gorman, Taurai Kambeu, Malvern Munjoma, Gabrielle O'Malley.

**Data curation:** Trina Gorman.

**Formal analysis:** Trina Gorman, Loida Erhard.

**Funding acquisition:** Trina Gorman.

**Investigation:** Trina Gorman.

**Methodology:** Trina Gorman, Bernardo Hernandez, Geoff Garnett, Loida Erhard, Taurai Kambeu, Malvern Munjoma, Gabrielle O'Malley.

**Project administration:** Trina Gorman, Taurai Kambeu, Malvern Munjoma, Brian Maponga, Sinokuthemba Xaba, Getrude Ncube, Jabulani Mavudze.

**Resources:** Trina Gorman, Taurai Kambeu, Brian Maponga.

**Software:** Trina Gorman, Loida Erhard.

**Supervision:** Bernardo Hernandez, Malvern Munjoma, Gabrielle O'Malley.

**Validation:** Trina Gorman.

**Visualization:** Trina Gorman.

**Writing – original draft:** Trina Gorman.

**Writing – review & editing:** Trina Gorman, Bernardo Hernandez, Geoff Garnett, Taurai Kambeu, Malvern Munjoma, Gabrielle O'Malley.

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
