## [Decision Letter · Decision Letter 0]

10 Oct 2024

PGPH-D-24-01412

Moving from results to action with community verification: A case study from a

performance based financing program in Zimbabwe

Dear Dr. Gorman,

Thank you for submitting your manuscript to PLOS Global Public Health. After careful consideration, we feel that it has merit but does not fully meet PLOS Global Public Health’s publication criteria as it currently stands. Therefore, we invite you to submit a revised version of the manuscript that addresses the points raised during the review process.

Please find the reviewers comments below and provide a detailed response to each of these.

We look forward to receiving your revised manuscript.

Kind regards,

Joanna Tindall, PhD

Staff Editor

Journal Requirements:

2. Please send a completed 'Competing Interests' statement, including any COIs declared by your co-authors. If you have no competing interests to declare, please state "The authors have declared that no competing interests exist". Otherwise please declare all competing interests beginning with the statement "I have read the journal's policy and the authors of this manuscript have the following competing interests:"

4. We ask that a manuscript source file is provided at Revision. Please upload your manuscript file as a .doc, .docx, .rtf or .tex.

5. We have noticed that you have uploaded Supporting Information files, but you have not included a list of legends. Please add a full list of legends for your Supporting Information files after the references list.

Additional Editor Comments (if provided):

Reviewers' comments:

Reviewer's Responses to Questions

**Comments to the Author**

1. Does this manuscript meet PLOS Global Public Health’s publication criteria ? Is the manuscript technically sound, and do the data support the conclusions? The manuscript must describe methodologically and ethically rigorous research with conclusions that are appropriately drawn based on the data presented.

Reviewer #1: Yes

Reviewer #2: Yes

2. Has the statistical analysis been performed appropriately and rigorously?

Reviewer #1: N/A

Reviewer #2: Yes

3. Have the authors made all data underlying the findings in their manuscript fully available (please refer to the Data Availability Statement at the start of the manuscript PDF file)?

Reviewer #1: Yes

Reviewer #2: Yes

4. Is the manuscript presented in an intelligible fashion and written in standard English?

Reviewer #1: Yes

Reviewer #2: Yes

5. Review Comments to the Author

Reviewer #1: Overall:

• This is a really important analysis and addresses a current gap in the literature. This paper will be helpful for other VMMC and HIV prevention programs to learn from particularly as programs plan for sustainability and transfer of services from outside funders to ministries of health.

ABSTRACT:

•Overall:

o It is unclear if this verification is for the national program and across all sites and regions or specific to a location or area or just Gates supported VMMC sites? Recommend specifying.

o Recommend updating ‘community verification’ to ‘patient verification’ because this introduces confusion to the reader because the verification is happening at the individual level and not the community which implies another type of verification. This remains unclear beyond the abstract with references to community verifiers and also references to verification of patients.

• Objectives:

o Recommend specifying which records are being verified with the patient records (i.e. MOH, national, regional, etc.)

o “the paths and processes that led to the results” – this could be removed since you don’t explain the “paths and processes” in the rest of the abstract.

• Methods:

o “For a VMMC program in Zimbabwe that took place from 2016-2018, we performed a mixed method verification including quantity verification, community verification, and verifier interviews to help understand the findings.” Recommend updating to improve clarity: “We performed a mixed method verification including quantity verification, community verification and interviews to verify records for the national (?) VMMC program from 2016 – 2018.”

o For the sentence: “We also assessed the correlation between the quantity and community verification performance scores to see whether facilities that have strong record keeping tended to also have strong validation from patients and vice versa”, recommend either removing this reference to performance scores since it isn’t clear how these scores were calculated in the abstract.

o Information is needed on how records were selected for this data verification exercise

• Results:

o “Of the 36,877 reported VMMCs, 94% were verified with facility records.” Were these nationally reported? If yes, please add to this sentence.

o For the sentence: “Community verifiers reported that some patients admitted providing incorrect contact information to avoid follow-up and most verifiers suspected staff had fabricated data.” Do you have any findings to support that staff fabricated data? Recommend not reporting that they ‘suspected’ which is not objective and presenting any data identified during the review to support that there was fabricated data.

• Conclusion:

o There are references to programs paying based on facility records, but these findings related to PBF aren’t mentioned in the results. The Conclusions should only present on information presented in the results otherwise it is unclear how these conclusions were drawn.

Main Paper:

• Methods:

o Study Setting: recommend adding sexual transmission to the last sentence “reduces the risk of female-to-male sexual transmission of HIV”

o PBF Program Design:

This section describes both PBF as supporting scale-up of VMMC but in the second sentence it refers to the activity as a ‘study’. Recommend specifying if this was a grant to support a study on the use of PBF to improve uptake of VMMC.

It should be specified if the PBF included the national program and both PEPFAR supported and MoHCC-supported sites or just Gates supported sites or a mix of all.

The sentence: “This study focused on VMMCs performed as part of a grant from May 2016 to April 2019, whereby PSI as well as facility staff were remunerated for each VMMC.” Were facility staff remunerated in addition to their regular pay? This should be clarified.

o Community verification:

Please specify how many attempts were made to contact VMMC clients before you considered the record ‘unverified’?

Specify age of minors in Zimbabwe

Was there a reason only Prepex device use was asked about and not ShangRing? Prepex use slowly decreased in Zimbabwe after the WHO report was released in 2016 indicating 2 doses of TTCV is required prior to prepex use. I believe Shangring was introduced in 2019 and likely overlapped with the period of interest. If this study was only conducted at Gates funded sites maybe Prepex use continued for longer?

It is confusing why verifiers had to rank their level of certainty about the potential for data fabrication when objective and quantifiable indicators were captured that would better indicate data fabrication.

• Results:

o For the sentence: “For the quantity verification, ninety-four percent (34,713/36,877) of selected records were sufficiently complete (C).” recommend quantifying what it means that the records were sufficiently complete.

o The description of the reasons verifiers suspected data fabrication is helpful but the indicators used to measure these suspicions are not standardized across reviewers but instead rely on reviewers individual experiences and interactions at the site level. The interpretation and reliability of these findings is limited. It is even mentioned that the verifiers didn’t have hard evidence about fabrication. There are substantial quantifiable results that should be the primary focus of the paper but these suspicions about data falsification take away from the validity and credibility of the other findings. Recommend removing with a mention in the discussion or improving the description of how these ‘suspicions’ were captured uniformly across sites in the Methods.

Reviewer #2: The article looks nice, you may only need to improve on the study design, it's not clear on the designs you employed. You used MEASURE Evalution verification tool, you need to indicate version and link to source. You can also add what you customized on the tool. May you add a table or figure that shows correlation test.

On qualitative results you need to add more information on how falsified information was handled.

6. PLOS authors have the option to publish the peer review history of their article (what does this mean? ). If published, this will include your full peer review and any attached files.

**Do you want your identity to be public for this peer review?** For information about this choice, including consent withdrawal, please see our Privacy Policy .

Reviewer #1: No

Reviewer #2: **Yes: ** Isaac Taramusi

---

## [Decision Letter · Decision Letter 1]

21 Feb 2025

PGPH-D-24-01412R1

Moving from results to action with community verification: A case study from a

performance based financing program in Zimbabwe

Dear Dr. Gorman,

Thank you for submitting your manuscript to PLOS Global Public Health. After careful consideration, we feel that it has merit but does not fully meet PLOS Global Public Health’s publication criteria as it currently stands. Therefore, we invite you to submit a revised version of the manuscript that addresses the points raised during the review process.

We look forward to receiving your revised manuscript.

Kind regards,

Annesha Sil, Ph.D.

Staff Editor

PLOS 

Reviewers' comments:

Reviewer's Responses to Questions

**Comments to the Author**

1. If the authors have adequately addressed your comments raised in a previous round of review and you feel that this manuscript is now acceptable for publication, you may indicate that here to bypass the “Comments to the Author” section, enter your conflict of interest statement in the “Confidential to Editor” section, and submit your "Accept" recommendation.

Reviewer #3: (No Response)

Reviewer #4: (No Response)

2. Does this manuscript meet PLOS Global Public Health’s publication criteria ? Is the manuscript technically sound, and do the data support the conclusions? The manuscript must describe methodologically and ethically rigorous research with conclusions that are appropriately drawn based on the data presented.

Reviewer #3: (No Response)

Reviewer #4: Partly

3. Has the statistical analysis been performed appropriately and rigorously?

Reviewer #3: (No Response)

Reviewer #4: Yes

4. Have the authors made all data underlying the findings in their manuscript fully available (please refer to the Data Availability Statement at the start of the manuscript PDF file)?

Reviewer #3: Yes

Reviewer #4: Yes

5. Is the manuscript presented in an intelligible fashion and written in standard English?

Reviewer #3: Yes

Reviewer #4: Yes

6. Review Comments to the Author

Reviewer #3: Review feedback is given in word document track change.

Reviewer #4: Thank you for the opportunity to review this paper. I have provided comments to help the author(s) bring this paper to the level where it can merit publication. Specifically, I recommend addressing the points outlined in the detailed feedback provided below.

Title

I think you need to revise the title of the article, because it doesn't necessarily reflect what you're saying. In your article, the results of the community check were not used to take action, nor to punish possible fraud.

Method

Can you specify who did what in the article? Specify who made the collection tools, data processing and analysis, etc.

In your case, this survey was not carried out. Why not?

What about data saturation? What research design was used for the qualitative – phenomenological, grounded theory, case study, etc.?

Can you draw up a theoretical model of the community verification process?

How were the auditors chosen? On what basis were PSI and Ministry of Health auditors chosen? Can you include a table showing the number of verifiers per period and per subcontractor? How were the subcontractors chosen? How were the auditors paid?

Discussion

In your key words you talk about unintended consequences, you also say that voluntary medical male circumcisions is a taboo subject, what are the consequences of community verification on patients who didn't want this to be known in their community.

There is a great risk that the auditors have defrauded the surveys declared by telephone or at home, you'll have to put this as the limit of your work since you haven't checked in your turn whether the people who are declared as surveyed by the auditors really have been.

Translated with DeepL.com (free version)

7. PLOS authors have the option to publish the peer review history of their article (what does this mean? ). If published, this will include your full peer review and any attached files.

**Do you want your identity to be public for this peer review?** For information about this choice, including consent withdrawal, please see our Privacy Policy .

Reviewer #3: **Yes: ** khem

Reviewer #4: No

---

## [Decision Letter · Decision Letter 2]

30 Jun 2025

PGPH-D-24-01412R2

A Results to Action Framework for Community Verification: A case study from a Performance Based Financing program in Zimbabwe

Dear Dr. Gorman,

Thank you for submitting your manuscript to PLOS Global Public Health. After careful consideration, we feel that it has merit but does not fully meet PLOS Global Public Health’s publication criteria as it currently stands. Therefore, we invite you to submit a revised version of the manuscript that addresses the points raised during the review process.

The manuscript has been evaluated by a reviewer, and their comments are available below. The reviewer is largely satisfied with your paper but have some minor issues they would like to see addressed.

Could you please revise the manuscript to carefully address the concerns raised?

We look forward to receiving your revised manuscript.

Kind regards,

Jenna Scaramanga

Staff Editor

Journal Requirements:

Additional Editor Comments (if provided):

Reviewers' comments:

Reviewer's Responses to Questions

**Comments to the Author**

1. If the authors have adequately addressed your comments raised in a previous round of review and you feel that this manuscript is now acceptable for publication, you may indicate that here to bypass the “Comments to the Author” section, enter your conflict of interest statement in the “Confidential to Editor” section, and submit your "Accept" recommendation.

Reviewer #1: All comments have been addressed

2. Does this manuscript meet PLOS Global Public Health’s publication criteria ? Is the manuscript technically sound, and do the data support the conclusions? The manuscript must describe methodologically and ethically rigorous research with conclusions that are appropriately drawn based on the data presented.

Reviewer #1: Yes

3. Has the statistical analysis been performed appropriately and rigorously?

Reviewer #1: N/A

4. Have the authors made all data underlying the findings in their manuscript fully available (please refer to the Data Availability Statement at the start of the manuscript PDF file)?

Reviewer #1: Yes

5. Is the manuscript presented in an intelligible fashion and written in standard English?

Reviewer #1: Yes

6. Review Comments to the Author

Reviewer #1: There are some grammatical errors and inconsistencies that should be addressed during the final edits of the manuscript:

“Pearson Correlation coefficients” should be “Pearson correlation coefficients” (this should not be capitalized)

Consistency:

*Recommend harmonizing formatting of percentages (e.g., “94% (34,713 of 36,877)”).

*Remove inconsistencies in decimal places (e.g., 0.68 instead of .68).

The denominator in some percentages is unclear or inconsistent. For example, the authors report 9.5% (191/2010), but the earlier sentence says 151 people reported not being circumcised implying the remaining sample is 1,859, not 2,010.

Suggestion: Use consistent denominators and be explicit:

“Among the 1,859 patients who reported being circumcised…”

The authors could consider a table footnote or brief line in the text explaining that 151 were excluded for reporting not being circumcised.

7. PLOS authors have the option to publish the peer review history of their article (what does this mean? ). If published, this will include your full peer review and any attached files.

**Do you want your identity to be public for this peer review?** For information about this choice, including consent withdrawal, please see our Privacy Policy .

Reviewer #1: No

---

## [Decision Letter · Decision Letter 3]

28 Jul 2025

A Results to Action Framework for Community Verification: A case study from a Performance Based Financing program in Zimbabwe

PGPH-D-24-01412R3

Dear . Gorman,

We are pleased to inform you that your manuscript 'A Results to Action Framework for Community Verification: A case study from a Performance Based Financing program in Zimbabwe' has been provisionally accepted for publication in PLOS Global Public Health.

Best regards,

Julia Robinson

Executive Editor

Reviewer Comments (if any, and for reference):

Reviewer's Responses to Questions

**Comments to the Author**

1. If the authors have adequately addressed your comments raised in a previous round of review and you feel that this manuscript is now acceptable for publication, you may indicate that here to bypass the “Comments to the Author” section, enter your conflict of interest statement in the “Confidential to Editor” section, and submit your "Accept" recommendation.

Reviewer #1: All comments have been addressed

2. Does this manuscript meet PLOS Global Public Health’s publication criteria ? Is the manuscript technically sound, and do the data support the conclusions? The manuscript must describe methodologically and ethically rigorous research with conclusions that are appropriately drawn based on the data presented.

Reviewer #1: Yes

3. Has the statistical analysis been performed appropriately and rigorously?

Reviewer #1: N/A

4. Have the authors made all data underlying the findings in their manuscript fully available (please refer to the Data Availability Statement at the start of the manuscript PDF file)?

Reviewer #1: Yes

5. Is the manuscript presented in an intelligible fashion and written in standard English?

Reviewer #1: Yes

6. Review Comments to the Author

Reviewer #1: (No Response)

7. PLOS authors have the option to publish the peer review history of their article (what does this mean? ). If published, this will include your full peer review and any attached files.

**Do you want your identity to be public for this peer review?** For information about this choice, including consent withdrawal, please see our Privacy Policy .

Reviewer #1: No
